# Occurrence of Hepatoblastomas in Patients with Beckwith–Wiedemann Spectrum (BWSp)

**DOI:** 10.3390/cancers15092548

**Published:** 2023-04-29

**Authors:** Steven D. Klein, Madison DeMarchis, Rebecca L. Linn, Suzanne P. MacFarland, Jennifer M. Kalish

**Affiliations:** 1Division of Human Genetics, Children’s Hospital of Philadelphia, Philadelphia, PA 19104, USA; kleinsd@chop.edu (S.D.K.); demarchism@chop.edu (M.D.); 2Department of Pediatrics, Perelman School of Medicine, Philadelphia, PA 19104, USA; macfarlands@chop.edu; 3Department of Pathology and Lab Medicine, Perelman School of Medicine, Children’s Hospital of Philadelphia, Philadelphia, PA 19104, USA; linnr@chop.edu; 4Division of Oncology, Children’s Hospital of Philadelphia, Philadelphia, PA 19104, USA; 5Center for Childhood Cancer Research, Children’s Hospital of Philadelphia, Philadelphia, PA 19104, USA; 6Department of Genetics, Perelman School of Medicine, Philadelphia, PA 19104, USA

**Keywords:** hepatoblastoma, Beckwith–Wiedemann syndrome (BWS), methylation

## Abstract

**Simple Summary:**

Beckwith–Wiedemann syndrome (BWS) is an overgrowth and cancer predisposition disorder that is associated with increased risk of hepatoblastoma (HB), a liver tumor, and Wilms tumors (WTs), a kidney tumor. Patients with BWS are screened for these tumors during their childhood. In this study, we present 16 cases of BWS with HB, focusing on their molecular diagnosis and clinical phenotype. We obtained liver and matched HB samples from 8 of the 16 cases and tumor-only samples from 2 additional cases. By analyzing the molecular subtypes of these liver samples, we were able to stratify BWS-HB into three distinct groups, providing insight into the oncogenesis of HB. Based on our data, we suggest that organ-specific mosaicism is a major hurdle for genotype–phenotype correlation within BWS.

**Abstract:**

Patients with Beckwith–Wiedemann syndrome (BWS), an epigenetic imprinting disorder involving alterations in genes at the 11p15 chromosomal location, are predisposed to develop hepatoblastomas (HBs), which are rare embryonal liver tumors. Tumors can develop after a BWS diagnosis or, conversely, can be the presenting feature leading to a subsequent diagnosis. While HBs are the cardinal tumors of BWS, not all patients with the BWS spectrum will develop HBs. This observation has led to many hypotheses, including genotype-associated risk, tissue mosaicism, and tumor-specific second hits. To explore these hypotheses, we present the largest cohort of patients with BWS and HBs to date. Our cohort comprised 16 cases, and we broadened our sample size by searching the literature for all cases of BWS with HBs. From these isolated case studies, we amassed another 34 cases, bringing the total number to 50 cases of BWS-HB. We observed that paternal uniparental isodisomy (upd(11)pat) was the most common genotype, representing 38% of cases. The next most common genotype was IC2 LOM, representing 14% of cases. Five patients had clinical BWS without a molecular diagnosis. To investigate the potential mechanism of HBs in BWS, we analyzed normal liver and HB samples from eight cases and isolated tumor samples from another two cases. These samples underwent methylation testing, and 90% of our tumor samples underwent targeted cancer next-generation sequencing (NGS) panels. These matched samples provided novel insights into the oncogenesis of HBs in BWS. We found that 100% of the HBs that underwent NGS panel testing had variants in the *CTNNB1* gene. We further identified three distinct groups of BWS-HB patients based on epigenotype. We also demonstrated epigenotype mosaicism, where 11p15 alterations can differ between the blood, HB, and normal liver. In light of this epigenotype mosaicism, tumor risk assessment based on blood profiling may not be accurate. Therefore, universal screening is recommended for all patients with BWS.

## 1. Introduction

Hepatoblastomas (HBs) are the most common malignant liver tumors in children [1,2]. Some patients with HBs have also been found to have higher birthweights (LGA), and this is thought to represent the relationship between HB risk and some overgrowth and cancer predisposition syndromes [2,3]. Up to 15 percent of patients with HBs may have a cancer predisposition syndrome [4]. The true prevalence of cancer predisposition syndromes (CPSs) in patients with HBs is challenging to assess because HBs are rare. The McGill Interactive Pediatric OncoGenetic Guidelines (MIPOGG) were developed as a prediction tool to assess CPS risk [5]. There is growing recognition of the need for tumor surveillance and workup of pediatric solid tumors associated with CPSs [6,7]. The 2016 AACR Childhood Cancer Predisposition Workshop committee proposed a uniform screening approach for syndromes with a risk of cancer >1% and entailing the development of HBs and/or Wilms tumors (WTs), embryonal tumors that affects the kidney(s) [8]. Surveillance for HB risk has been suggested for patients with underlying disorders, including Beckwith–Wiedemann syndrome (BWS), Simpson–Golabi–Behmel syndrome, and familial adenomatous polyposis [8,9].

Beckwith–Wiedemann syndrome/spectrum (BWS) is the most common childhood epigenetic cancer predisposition disorder and describes the clinical spectrum of patients with (epi)genetic changes that affect 11p15 imprinting domains [10,11]. The underlying molecular changes that have been found in patients with BWS are complex, and epigenetic mosaicism presents a challenge for accurate molecular characterization. The most common causes include loss of methylation of the *KCNQ1OT1* transcription start site differentially methylated region (*KCNQ1OT1:TSS* DMR), also known as imprinting control region 2 (IC2 LOM), and paternal uniparental isodisomy for chromosome 11 (upd(11)pat). Less commonly, gain of methylation at the H19/IGF2 intergenic differentially methylated region (H19/IGF2:IG DMR), also known as imprinting control region 1 (IC1 GOM), is observed [10]. Finally and least commonly, copy number alterations, translocations, or mutations in CDKN1C may be observed [10].

The classic BWS phenotype includes macrosomia, omphalocele, macroglossia and body asymmetry. Since the reclassification of BWS as a spectrum, a growing number of patients with atypical BWS phenotypes have been identified through positive molecular testing [11]. One of the groups of patients with atypical phenotypes comprises those who are found to have molecular testing results consistent with a BWS-related change at the time of tumor diagnosis and are then recognized to have features and/or clinical histories consistent with BWS [11,12,13].

The risk for HBs has been linked to the BWS population from early characterization and epidemiologic studies [14,15]. Patients with BWS have a 2280 relative risk for HBs during the first four years of life compared to the general pediatric population, which appears to differ in association with the BWS molecular subtypes based on blood testing [10,15,16,17]. This association demonstrated an increased risk for WTs with IC1 GOM, leading to the hypothesis that IC1 alterations are more common in WTs [18]. Reproducible studies have demonstrated a lower relative tumor risk (both WTs and HBs) associated with IC2 LOM compared to IC1 GOM and upd(11)pat [17,19]. The observation of HBs associated with IC2 LOM or upd(11)pat led to the suggestion that IC2 may influence HB development [20]. It was further suggested that patients with the IC2 LOM profile may experience a risk for HBs due to the fetal growth patterns associated with this molecular subtype [16,21].

## 2. Materials and Methods

### 2.1. Patient Cohort

Patients with an HB diagnosis who received comprehensive workup for 11p15 mosaicism between 2015 and 2022 at the Children’s Hospital of Philadelphia were included under IRB-approved protocols (IRB13-010658, IRB19-016459,and IRB19-016677). Patients were evaluated by genetics, oncology, and pathology, and clinical features were abstracted from the medical record as previously described [11,13]. We queried the literature for additional cases by searching for “Beckwith-Wiedemann Syndrome” and “Hepatoblastoma” on 25 January 2023, which yielded 132 references (Appendix A). From these 132 papers, we searched for case reports that provided clinical, molecular, and/or pathological data. We found 35 cases from the literature search. These case reports were reviewed for genotype, age at diagnosis, sex, pathology, tumor size, treatment course, and recurrence.

### 2.2. Molecular Testing

11p15 methylation testing was performed in the Genetic Diagnostic Laboratory (GDL) at the University of Pennsylvania and the mosaic burden was calculated as previously described [22]. Cases with uniparental disomy were confirmed with a chromosomal microarray performed with either blood, tissue, or tumor samples at the Children’s Hospital of Philadelphia (this test can detect mosaicism down to 1% [23]). Solid tumor panel testing was performed at the Children’s Hospital of Philadelphia in three formats. The majority (seven) of the patients underwent the Comprehensive Solid Tumor Panel V2.0, which is a test that analyzes 238 cancer genes and approximately 600 fusion genes. The test uses next-generation sequencing and data analysis to sequence all the coding exons and flanking intron sequences of targeted genes in the panel, and select known intronic mutations are also evaluated. The test also evaluates copy number variations for gross deletions and duplications using NGS data. The results are reported based on the HGVS nomenclature and classified into tier 1–3 variants [24]. An additional two patients underwent Comprehensive Solid Tumor Panel V2.0 with the addition of paired normal tissue testing. One patient underwent the Solid Tumor Panel, which is comparable to the comprehensive panel but without fusion analysis.

## 3. Results

### 3.1. Demographics, Epigenotypes, and Pathologic Types of Patients with BWSp-HB

We included our 16 patients along with 35 cases from the literature, resulting in a total of 51 BWS-HB cases for analysis. Among the 51 cases, there was a slight female predominance, with 59 percent of the patients being female. The age of diagnosis ranged from in utero to 22 years in the literature (mean: 15.54 months, median: 6.5 months) and birth to 25 months in our cohort (mean: 5.7 months, median: 4 months), highlighting the importance of early screening for patients with BWS (Table 1). DNA epigenotyping from blood was available in 35 of the 51 BWS-HB cases (Figure 1). We found that the two most common genotypes were upd(11)pat and IC2 LOM. We included one patient from the literature with upd(11)pat who also carried a paternally inherited frameshift mutation in the *ABCC8* gene (c.3512delT; p.L1171fs) (patient 33, [25]); the latter was thought to be causative of hyperinsulinism. In the literature, there were cases reported with some non-classic BWS genotypes, including loss of heterozygosity at the calcitonin and insulin loci on the short arm of chromosome 11 (patient 17 [26]), deletion within IC1 (patient 20, [27]), and interstitial deletion of both 11p11.1 and part of 11p11.2 in all cells, with complete inversion of the heterochromatic block of one chromosome 9 (patient 28, [28]). Notably, there were no cases identified with IC1 gain of methylation (GOM) in our study or in the literature. HB pathology was available/reported in 30 of the 50 HB-BWS cases, revealing that all hepatoblastomas had a mixed epithelial histology (100%), with only a small percentage having mesenchymal or cholangioblastic components. These findings are consistent with previous reports that have shown that the epithelial subtype is the most common histological HB subtype [29]. Mesenchymal components are characterized by the presence of primitive spindle-shaped cells that can differentiate into various types of tissue, such as osteoid, muscle, and/or fat, while cholangioblastic components are characterized by the presence of bile duct-like structures. These subtypes are less common in BWS-HBs [30,31]. Notably, we found no small cell undifferentiated (SCU) components for any tumor in our cohort or the literature.

### 3.2. BWS Clinical Features

We next evaluated the prevalence of classic clinical features of BWS in patients with BWS-HBs. Many of the patients had classic features of BWS (Table 2). More than 70 percent of the patients had lateralized overgrowth, macroglossia, ear creases/pits, and/or nevus simplex. More than 50 percent of the patients had hypoglycemia/hyperinsulinism. Interestingly, less than 50 percent of the patients in our study had omphalocele, nephromegaly, and/or hepatomegaly (Table 2). We present the clinical scores for the 16 patients in our cohort in Table 3.

### 3.3. Normal Liver and Tumor Methylation Analysis

We examined the molecular profiles of normal tissue and tumor samples from patients with BWS-HBs. For the 16 patients in our study, we obtained results for normal tissue and tumor samples for 8 patients and isolated tumor results without matched normal liver results for an additional 2 patients. We first examined whether the genotype in the blood matched that in the liver and the tumor (which we refer to as a consistent genotype) or differed (which we refer to as an inconsistent genotype). A consistent epigenotype refers to a stable epigenetic modification that is present in all cells of an individual’s body and maintained throughout development and aging. An inconsistent epigenotype, on the other hand, refers to a dynamic or unstable epigenetic modification that is present in only some cells of an individual’s body or varies over time. Our findings revealed five cases of inconsistent genotypes (case IDs 10, 13, 14, 15, and 16), highlighting the phenomenon of multiple methylation statuses being present in a single patient. In case ten, the blood genotype showed IC2 LOM, while the normal liver and HB showed both IC2 LOM and IC1 GOM without upd(11)pat occurring. In cases 15 and 16, there were no abnormalities in the blood, but the normal livers showed IC1 GOM and the HBs showed loss of heterozygosity resulting in upd(11)pat. Finally, in case 14, no testing of blood was carried out, but the normal liver showed IC1 LOM and the HB showed loss of heterozygosity resulting in upd(11)pat. These findings demonstrate that the potential for multiple distinct and inconsistent epigenotypes exists in BWS livers and HBs (Table 3).

### 3.4. Solid Tumor Panel Analysis

We analyzed the results of the solid tumor panels performed for ten individual tumors, including two tumors from patient one. All ten solid tumor panels showed mutations in the gene CTNNB1, a known driver gene in the pathogenesis of HBs. CTNNB1, which encodes beta-catenin, plays a critical role in the Wnt signaling pathway, which regulates cell growth and differentiation. Many of the mutations that we observed were in “hotspots”, including serine 33 (case 1), glycine 34 (cases 1 and 4), and aspartate 32 (case 14).

## 4. Discussion

### 4.1. Risk Stratification Based on Blood Genotypes in BWS-HBs

Our cohort demonstrates that upd(11)pat and IC2 LOM are the most common genotypes in patients with BWS-HBs. Notably, we did not find any cases with IC1 GOM alone in the tumor in our cohort or in the literature. These findings strongly suggest that different genotypes may contribute to distinct molecular pathways leading to the development of HBs in patients with BWS. While our data support the increased risk of HBs in patients with upd(11)pat and IC2 LOM, we also observed the occurrence of HBs in patients with normal methylation testing results from blood. Further analysis of these cases revealed that three patients had IC1 GOM in unaffected livers, with loss of heterozygosity resulting in upd(11)pat in the HBs. These patients had lower BWS clinical scores and only showed a few classic BWS features, such as macroglossia, lateralized overgrowth, and ear creases. Our cohort highlights that there can be both mosaicism within a given genotype and multiple distinct epigenetic/genetic events leading to dysregulation of the 11p15 critical region in patients with BWS and HBs [30]. We propose that this mechanism hinges on both cell burden and clonal mosaicism (Figure 2).

### 4.2. Two Distinct Groups of BWS-HBs

We identified two distinct groups of BWS-HBs based on the underlying epigenotype. These groups provide valuable insights into the molecular mechanisms that underlie the development of HBs in patients with BWS. Group one, or the consistent genotype, is defined by the detection of upd(11)pat or IC2 LOM changes in blood, which in some cases are ubiquitous and also present in the unaffected liver and the HB. This group represents the most common epigenotype in patients with BWS-HBs and the mosaicism model (Figure 2) [31,32].

Group two, or the inconsistent genotype, is defined by discrepant tissue genotypes. Within this group, we identified one case where IC2 LOM changes were detected in blood, with additional IC1 GOM changes then observed in the affected liver and in the HB without upd(11)pat. Chromosome microarray analysis determined that this was not upd(11)pat. Furthermore, within the discrepant group, we identified a case with normal testing results from blood and IC2 LOM in the HB and three cases with normal testing results from blood, IC1 GOM in the normal liver, and loss of heterozygosity resulting in upd(11)pat in the HB. Example of the changes in the latter group (patient 14) included 68% methylation at IC1 and 50% methylation at IC2 in the normal liver (IC1 GOM). In the HB, IC1 methylation increased to 79% and IC2 methylation decreased to 41%, with SNP array analysis showing loss of heterozygosity/upd(11)pat in the HB. These groups underscore the important and likely critical role of the IC2 region in the pathogenesis of HBs in patients with BWS, as 100% of the HBs from our study showed alterations at IC2 leading to IC2 LOM. Our results suggest that LOM at IC2 is likely a key event that leads to cancer predisposition and, ultimately, HB formation or at least creates the permissive environment. LOM at this region leads to loss of the maternally expressed cyclin-dependent kinase inhibitor 1C (CDKN1C) and potassium voltage-gated channel subfamily Q member 1 (*KCNQ1*) and gain of expression of potassium voltage-gated channel subfamily Q member 1 long non-coding RNA (*KCNQ1OT1*). *CDKN1C* is a maternally expressed imprinted gene located in the IC2 region on chromosome 11p15.5. It encodes a protein known as p57KIP2, which is a cyclin-dependent kinase inhibitor involved in the regulation of cell cycle progression [15,33,34]. It is well-established that the proper regulation of cell cycle progression is critical for maintaining genomic stability and preventing the development of cancer. Loss of *CDKN1C* function can disrupt these checkpoints and allow cells to bypass critical mechanisms that prevent uncontrolled cell growth. When mutated cells lose these regulatory mechanisms, they can evade apoptosis, a process that normally eliminates damaged or abnormal cells, and continue to proliferate, leading to the formation of a tumor [35]. Understanding the role of *CDKN1C* in the regulation of cell cycle checkpoints and its relationship with cancer development is crucial for the development of targeted therapeutic strategies.

### 4.3. The Role of CTNNB1 in BWS-HBs

The presence of *CTNNB1* mutations in 100% of the BWS-HB tumors tested reaffirms that, while it is likely a later step in BWS-HB transition, *CTNNB1* is clearly part of this pathogenesis. The recurrence of mutations at the hotspot residues serine 33 (case 1), glycine 34 (cases 1 and 4), and aspartate 32 (case 14,) within our cohort, as well as the known involvement of *CTNNB1* in HB oncogenesis, highlights the likelihood that they are oncogenic drivers. Perhaps the most convincing case from the cohort exemplifying this was case one, where the patient presented with two distinct HBs. Each one had a separate *CTNNB1* mutation, both at hotspots but with distinct changes. This showed that (1) neither mutation was germline or in the unaffected liver and (2) these mutations were likely necessary but later steps in HB formation. Supported by our previous transcriptomics work, our current model is that liver overgrowth and cancer predisposition are driven by dysregulation of IC2 on 11p15 [32]. This environment either supports or cannot contain *CTNNB1* mutation events, allowing for clonal expansion of cells, which leads to HBs. More mechanistic work is needed to understand the direct link between IC2 dysregulation and its causality in *CTNNB1* mutation events. From analysis of the hotspot mutations, it is known that they cluster around a critical binding site for glycogen synthase kinase 3 beta (GSK3B) [33]. GSK3B may further represent a signaling node between Wnt, hedgehog, and mTOR signaling [34].

## 5. Conclusions

Our study characterized the underlying genotypes in a large cohort of patients with BWS and HBs. We confirmed previous observations that upd(11)pat and IC2 LOM are the most common epigenotypes in patients with BWS-HBs. For some patients, we reported consistent epigenotypes across the blood, liver, and hepatoblastoma, suggesting that oncogenesis is based on cell-burden mosaicism. In addition, we identified a subset of patients with inconsistent epigenotypes. These patients had normal blood epigenotypes but showed IC1 GOM in the unaffected liver, with additional loss of heterozygosity that appeared similarly to upd(11)pat in the HB. This stepwise clonal event is distinct from the consistent epigenotype group but ultimately culminates in the same result: dysregulation of IC2 and, specifically, the loss of expression of *CDKN1C* and *KCNQ1* and gain of expression of *KCNQ1OT1*. We propose that the lack of *CDKN1C* in the liver creates a permissive environment where mutations in *CTNNB1* can evade normal regulatory mechanisms. This is supported by the detection of *CTNNB1* mutations in every tumor that was tested. To gain further insight into the molecular mechanisms involved, we identified recurrent hotspot mutations in CTNNB1, which clustered around the binding site for GSK3B. This implicates the Wnt, mTOR, and hedgehog signaling pathways as potential targets for therapeutic intervention. Our cohort highlights that there can be both mosaicism within a given genotype [35] and multiple distinct epigenetic/genetic events leading to dysregulation of the 11p15 critical region in patients with BWS and HBs. We believe that our study provides a more refined understanding of the temporal mechanism of HB development in BWS, which may ultimately lead to the development of more effective therapeutic strategies. Specifically, we propose that targeting the dysregulated pathways—particularly, the Wnt, mTOR, and hedgehog pathways—could be a promising avenue for arresting the neoplastic transition and preventing the development of HBs in these patients. Our findings also confirm that there are limitations when using blood profiling as a basis for tumor risk assessment. We propose that universal screening for HBs should be considered in all patients with clinical BWS, regardless of their genotype in blood, because of the potential for divergent genotypes and to provide timely and individualized care and screening. Our work also provides insights into the molecular and clinical characteristics of BWS-HBs.

## Figures and Tables

**Figure 1 cancers-15-02548-f001:**
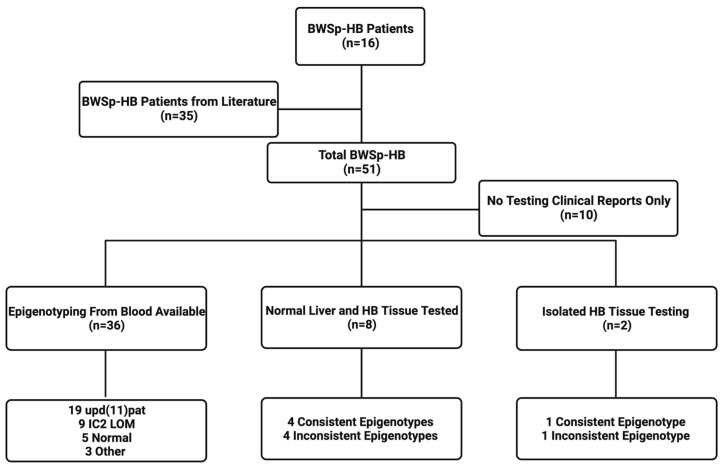
Flow sheet summarizing data collection for the BWS-HB study. The figure shows a flow sheet summarizing the data collection process for the Beckwith–Wiedemann spectrum-associated hepatoblastoma (BWS-HB) study. The flow sheet includes the following steps: cohort formation, addition of cases from the literature, and epigenotype analysis from the blood, tumor, and normal liver. A consistent epigenotype refers to a stable epigenetic modification that is present in all cells of an individual’s body and maintained throughout development and aging. An inconsistent epigenotype, on the other hand, refers to a dynamic or unstable epigenetic modification that is present in only some cells of an individual’s body or varies over time.

**Figure 2 cancers-15-02548-f002:**
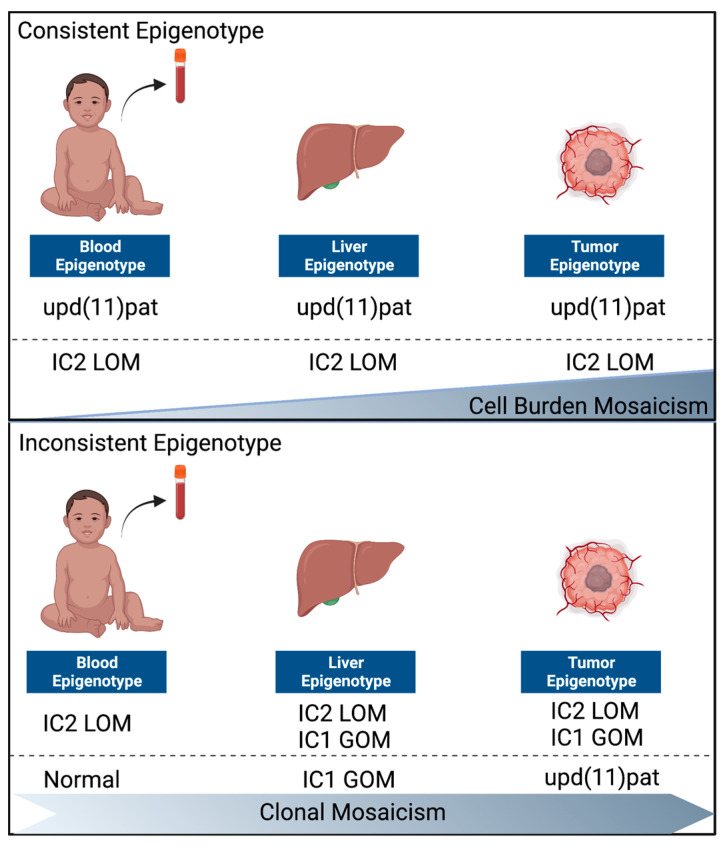
Comparison of consistent and inconsistent epigenotypes in Beckwith–Wiedemann syndrome-associated hepatoblastomas (BWS-HBs). Consistent epigenotypes are characterized by stable and heritable epigenetic modifications that are present in all tissues of an individual’s body and maintained throughout development and aging. Inconsistent epigenotypes, on the other hand, are characterized by dynamic or unstable epigenetic modifications that are present in only some cells of an individual’s body or vary over time.

**Table 1 cancers-15-02548-t001:** Demographics, epigenotypes from blood, and pathologic types of patients with Beckwith–Wiedemann syndrome-associated hepatoblastomas (BWS-HBs).

	This Report	The Literature	Total
Total	16	35	51
Male	5 (31%)	16 (46%)	21 (41%)
Female	11 (69%)	19 (54%)	30 (59%)
Age of diagnosis min	Birth	in utero	in utero
Age of diagnosis max	25 months	22 years	22 years
Epigenotype from Blood	15	20	35
upd(11)pat	7 (47%)	13 (65%)	20 (57%)
IC2 LOM	5 (33%)	4 (20%)	9 (26%)
Normal	3 (20%)	n/a	3 (9%)
Other	0	3 (15%)	3 (9%)
Pathology	14	16	30
Mixed epithelial	14 (100%)	16 (100%)	30 (100%)
+Mesenchymal	4 (29%)	1 (6%)	5 (17%)
+Cholangioblastic	1 (7%)	1 (6%)	2 (7%)

The table summarizes the following information for the cohort of new patients and those from the literature: sex, age of diagnosis, and epigenotype from blood (e.g., loss of methylation at KCNQ1OT1:TSS-DMR, gain of methylation at H19/IGF2:IG-DMR). The pathologic type of HB (e.g., mixed epithelial, mesenchymal, cholangioblastic) is also indicated.

**Table 2 cancers-15-02548-t002:** Clinical features of patients with BWS-HBs in our cohort.

Clinical Features	# of Patients (n)	%
Lateralized overgrowth	14 (16)	88%
Nevus simplex	8 (10)	80%
Macroglossia	11 (15)	80%
Ear creases	8 (10)	73%
Umbilical hernia/diastasis recti	5 (8)	55%
Hypoglycemia	8 (15)	53%
Hyperinsulinism	5 (13)	38%
Placental mesenchymal dysplasia	1 (3)	33%
Omphalocele	3 (13)	23%
Nephromegaly	3 (13)	21%
Hepatomegaly	2 (13)	15%

The table displays the clinical features of patients with BWS-HBs. The number of patients (n) and the percentage (%) with each clinical feature are shown.

**Table 3 cancers-15-02548-t003:** Blood, liver, and hepatoblastoma (HB) epigenotype results and *CTNNB1* mutation status for BWS-HBs.

ID	BWS Clinical Score	BWS Blood Results	BWS Liver Results	BWS HB Results	*CTNNB1* Mutation
1	9	upd(11)pat	upd(11)pat	upd(11)pat	c.101G>T, c.98C>T
2	10	upd(11)pat	upd(11)pat	upd(11)pat	c.99_101dup
3	8	upd(11)pat	\	upd(11)pat	c. 65_100del
4	10	upd(11)pat	\	\	c.100G>A
5	10	upd(11)pat	\	\	\
6	10	upd(11)pat	\	\	\
7	3	upd(11)pat	\	\	\
8	12	IC2 LOM	IC2 LOM	IC2 LOM	\
9	6	IC2 LOM	IC2 LOM	IC2 LOM	c.85_102del
10	5	IC2 LOM	IC2 LOM/IC1 GOM	IC2 LOM/IC1 GOM	c.122C>T
11	6	IC2 LOM	\	\	\
12	10	IC2 LOM	\	\	\
13	3	Normal	\	IC2 LOM	c.14-28_125del
14	2	\	IC1GOM	upd(11)pat	c.87_241+71del
15	4	Normal	IC1 GOM	upd(11)pat	c.95A>T
16	6	Normal	IC1 GOM	upd(11)pat	\

The table provides a summary of the blood, liver, and hepatoblastoma (HB) epigenotype results and the *CTNNB1* mutation status for each BWS-HB. The BWS blood results for each patient are indicated and can include the following: upd(11)pat—paternal uniparental disomy of chromosome 11, IC2—loss of methylation at the *KCNQ1OT1:TSS-DMR*, IC2/IC1—partial loss of methylation at *KCNQ1OT1:TSS-DMR* and gain of methylation at H19/IGF2:IG-DMR, IC1—gain of methylation at the *H19/IGF2:IG-DMR*, and normal—no epigenetic abnormalities detected.

## Data Availability

The data presented in this study are available on request from the corresponding author.

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
