# Peer review of "Occurrence of Hepatoblastomas in Patients with Beckwith–Wiedemann Spectrum (BWSp)"

_cancers, 2023, doi:10.3390/cancers15092548_

Round 1

Reviewer 1 Report

In the paper, the authors analyze a large cohort of patients with HB and BWS. The work is very interesting and could be significant for cancer screening and surveillance in BWS children, regardless of genotype. Moreover, the correlation between HBs in BWS and the presence of somatic mutations in CTNNB1 gene is a novelty and could be promising for possible pharmacological treatment, as suggested by the authors.

However, there are a couple of points that should be clarified:

  • In my opinion, we cannot define “germline” epigenotypes, because they are usually somatic and postzygotic events. Are the authors’ cases 0% of methylation?
  • Please, the authors should explain why they exclude that the HB’s molecular defect of patient 10 is a UPD. How the UPD was assessed? An explicative figure or a table may be helpful.
  • The authors should specify that patients 13-16 who test normal blood may have undetectable low-grade mosaicism in this tissue. Were other tissues, commonly used for diagnosis (such as buccal swabs) tested?
  • What do the authors mean by "IC1 GOM progress in UPD11pat"? Since these are two distinct genetic mechanisms, which do the authors hypothesize?
  • Molecular testing section: Have the authors used a commercial panel to characterize HBs? Please specify. If it is a custom panel, the authors could insert the list of analyzed genes.
  • Given the relevance of this study, the raw data should be deposited.
  • The definitions of “consistent” and “Inconsistent” epigenotypes may be more easily expressed, respectively, as cellular mosaicism and tissue mosaicism.

Minor points:

  • Lines 69-72: the extended name of the DMRs should be reported.
  • Lines 109-110: Please check the sentence.
  • Line 125 and Table 1: in the text, the authors report: “age of diagnosis ranged from in utero to 22 years”, but Table 1 reports “birth to 25 months”. Please check the data and make them concordant.
  • Table 1 and lines 150-155: The caption of the table does not refer to the table. Please check the data.
  •  Lines 136-137: Please, check the paragraph.
  • Lines 139-141: Please add a reference.
  • Lines 187-190: please move this sentence in the text.
  • Line 264: “all cells” should be replaced with “all tissues”.
  • Lines 222-227: this paragraph could be moved to the conclusions
  • Tables: I would recommend toning down the dark gray of the two tables, which makes the text difficult to read
  • Table 1 Row 11: The total number is incorrect.
  • Figure 1: Please check the data of the flow chart. The reported numbers are not concordant with the text.
  • Figure 1: 8 IC2 LOM, in place of 8 LC2 LOM
  • Table 3: Check the italics.
  • Table 3: uniform the variants’ nomenclature.
  • Table 3 row 11, 3rd column: add IC1 GOM.
  • Sheet#2 of the Supplementary file is never mentioned in the text. Moreover, it reports 35 patients, not 50 as indicated in the study.

Author Response

Dear Reviewer 1,

We would like to express our appreciation for your time and effort in reviewing our manuscript and providing us with valuable feedback. Your insightful comments have helped us improve the quality and clarity of our work.

We have carefully considered your suggestions and made the following changes to our manuscript:

Major Points –

In my opinion, we cannot define “germline” epigenotypes, because they are usually somatic and postzygotic events. Are the authors’ cases 0% of methylation?

Point One: We thank you for your feedback regarding the definition of germline epigenotype. Based on your comment, we have revised the manuscript and removed the word germline from the discussion of epigenotypes. We replace this with “epigenotype from blood” which we believe represents a good proxy for a large collection of cells that are widely distributed throughout the bone marrow. The methylation testing used is sensitive for mosaicism done to 2.5 %. Negative testing results mean the results are below this threshold.

Please, the authors should explain why they exclude that the HB’s molecular defect of patient 10 is a UPD. How the UPD was assessed? An explicative figure or a table may be helpful.

Point Two: We appreciate your question regarding patient 10 and UPD. This patient does not have UPD based on chromosomal microarray analysis (which has a sensitivity for detecting mosaicism down to 1%), we have added this clarification to the methods section.

The authors should specify that patients 13-16 who test normal blood may have undetectable low-grade mosaicism in this tissue. Were other tissues, commonly used for diagnosis (such as buccal swabs) tested?

Point Three: We appreciate your point regarding the use of buccal swabs for methylation testing. We have not seen an increased yield of positive results in buccal samples compared to blood, and therefore do not typically collect these samples. Our second line tissue for testing includes skin biopsies or other affected tissues such as tongue, pancreas, liver, or kidney where available. Skin biopsies were not performed on patients 13-16.

What do the authors mean by "IC1 GOM progress in UPD11pat"? Since these are two distinct genetic mechanisms, which do the authors hypothesize?

Point Four: We thank you for bringing to our attention the potential confusion in the wording of the sentence regarding patients with normal blood epigenotypes. We have revised the sentence to provide greater clarity and accuracy. The sentence now reads “These patients have normal blood epigenotypes but show IC1 GOM in the unaffected liver, with additional loss of heterozygosity that appears like upd(11)pat in the HB.”

Molecular testing section: Have the authors used a commercial panel to characterize HBs? Please specify. If it is a custom panel, the authors could insert the list of analyzed genes.

Point Five: We appreciate your comment and have included a detailed description of the tests used in the methods section, including their availability and commercial sources. The section now reads “11p15 methylation testing was performed in the Genetic Diagnostic Laboratory (GDL) at the University of Pennsylvania and the mosaic burden was calculated as previously described [22]. Cases with uniparental disomy were confirmed with a chromosomal microarray performed on either blood, tissue, or tumor sample performed at the children’s hospital of Philadelphia (This test can detect mosaicism down to 1%[23] (Conlin et al Hum Mol Gen 2010). Solid tumor panel testing was performed at the Children’s Hospital of Philadelphia in three formats. The majority (7) of the patients underwent the Comprehensive Solid Tumor Panel V2.0 which is a test that analyzes 238 cancer genes and approximately 600 fusion genes. The test uses next generation sequencing and data analysis to sequence all coding exons and flanking intron sequences of targeted genes in the panel and select known intronic mutations are also evaluated. The test also evaluates copy number variation analysis for gross deletions and duplications using NGS data. The results are reported based on the HGVS nomenclature and classified into tier 1-3 variants[23]. An additional two patients underwent Comprehensive Solid Tumor Panel V2.0 with the addition of paired normal tissue testing.  One patient had the Solid Tumor Panel which is comparable to the comprehensive panel without fusion analysis.”

Given the relevance of this study, the raw data should be deposited.

Point Six: We appreciate your comment regarding the raw data for this study. The genetic testing results are from the clinical testing reports and there are no additional data to deposit.

The definitions of “consistent” and “Inconsistent” epigenotypes may be more easily expressed, respectively, as cellular mosaicism and tissue mosaicism.

Point Seven: We appreciate your comment regarding the nomenclature used for cellular and tissue mosaicism.  While we also understand what the reviewer means by these terms, we think that the reader may be confused by these designations. We opted for “consistent” and “inconsistent” as they are more descriptive of the data without needing expertise in clonal mosaicism for understanding of the key point.

Minor points have been addressed in the text with tracked changes, and we thank you for your attention to detail.

Once again, we appreciate your time and feedback and look forward to hearing from you again.

Sincerely,

Steven Klein and Jennifer Kalish

Reviewer 2 Report

In this study, the authors report the clinical and molecular features of a cohort of patients affected by Beckwith-Wiedemann syndrome (BWS) and hepatoblastoma. They included 16 new cases and another 34 cases derived from literature, which is the largest cohort of BWS with hepatoblastoma reported so far. They analyzed the epi-genotype of chrm 11p15 and reported that the two most common molecular abnormalities in blood DNA were upd(11)pat and IC2 LOM. They also tested the 11p15 epi-genotype in normal and neoplastic liver and found several cases in which the molecular defect detected in blood or normal liver did not match that found in the tumor indicating epigenetic mosaicism. Finally, panel analysis on ten tumor samples demonstrated CTNNB1 mutations in all tested cases.

The study is interesting and deserves publication. I have a few comments that may improve the manuscript:

1.     The authors observed a lower BWS clinical score in the patient group with inconsistent epigenotype. Please include in Table 3 the phenotypic features and clinical scores of the 16 BWSp-HB cases analyzed.

2.     It would be interesting to know the DNA methylation level detected in the different tissues analysed to see if the 11p15 molecular defect was selected in cancer cells. In particular, what is the DNA methylation levels of IC1 and IC2 in the tumors with inconsistent epigenotype?

3.     It would be nice to see more information of the identified CTNNB1 variants.

Minor ponts:

1.     There is something missing in the sentence 109-111.

2.     The range of age of diagnosis reported in lines 125-127 is not the same of that reported in Table 1.

Author Response

Dear Reviewer 2,

Thank you for your time and effort in reviewing our manuscript and providing us with helpful feedback to  improve the quality and clarity of our work.

We have carefully considered your suggestions and made the following changes to our manuscript:

Major Points –

The authors observed a lower BWS clinical score in the patient group with inconsistent epigenotype. Please include in Table 3 the phenotypic features and clinical scores of the 16 BWSp-HB cases analyzed.

Point One: We have incorporated the BWS clinical scores into Table 3.

It would be interesting to know the DNA methylation level detected in the different tissues analysed to see if the 11p15 molecular defect was selected in cancer cells. In particular, what is the DNA methylation levels of IC1 and IC2 in the tumors with inconsistent epigenotype?

Point Two: We have added this detail for patient 14: “An example of the changes in the latter group (patient 14) includes 68% methylation at IC1 and 50% methylation at IC2, in normal liver (IC1 GOM). In the HB, IC1 methylation increases to 79% and IC2 methylation decreases to 41%, with SNP array showing loss of heterozygosity/upd(11)pat in the HB.”

It would be nice to see more information of the identified CTNNB1 variants.

Point Three: We have included the cDNA location for each variant, explained the recurrent hotspot mutations, and provided insight into the potential functional role of GSK3B in this context.

Minor points have been addressed in the text with tracked changes, and we thank you for your attention to detail. We believe that these changes have significantly strengthened the manuscript.

Once again, we appreciate your time and feedback and look forward to hearing from you again.

Sincerely,

Steven Klein and Jennifer Kalish 

Reviewer 3 Report

This is a well planned and executed retrospective analysis which is confirmatory of the current approach to tumor screening in Hb. The sample size preckuded further analyses which would have been of potential interest (correlations between genotype and known Hb risk factors such as AFP at diagnosis, PRETEXT, etc...). A future analysis including such correlations would be of great value.

Author Response

Dear Reviewer 3,

We would like to express our appreciation for your time and effort in reviewing our manuscript and providing us with valuable feedback. We agree that a future paper with more details relating relationship between genotype and known risk factors for HB, such as alpha-fetoprotein (AFP) levels at diagnosis and PRETEXT scores would be of great interest and value to the field.

Once again, we appreciate your time and feedback,

Sincerely,

Steven Klein and Jennifer Kalish